# VCRL: Variance-based Curriculum Reinforcement Learning for Large Language Models

## Abstract

Policy-based reinforcement learning currently plays an important role in improving LLMs on mathematical reasoning tasks. However, existing rollout-based reinforcement learning methods (GRPO, DAPO, GSPO, etc.) fail to explicitly consider LLMs' learning ability for samples of different difficulty levels, which is contrary to the human cognitive process of mathematical reasoning tasks from easy to difficult. Intuitively, we find that the variance of the rollout group's reward in Reinforcement Learning with Verifiable Rewards (RLVR) partly reflects the difficulty of the current sample for LLMs. Samples that are too easy or too difficult have a lower variance, while samples with moderate difficulty have a higher variance. Based on this, we propose VCRL, a curriculum reinforcement learning framework that dynamically controls the difficulty of training samples based on the variance of group rewards. Experiments on five mathematical benchmarks and two models reveal the advantages of VCRL over the current LLM RL baselines. Code is available at `https://anonymous.4open.science/r/VCRL-BD7E`.

## 1 Introduction

The new generation of large language models (LLMs) that use long Chain-of-Thoughts (CoTs) for reasoning (Xu et al., 2025a) have achieved remarkable results in information extraction (Zhang et al., 2025d; Jiang et al., 2024; 2025a), mathematics (Wang et al., 2025a), code (Yang et al., 2025b), and agent (Gao et al., 2025; Zhang et al., 2025b) fields, including GPT-5[1], GPT-OSS (Agarwal et al., 2025), DeepSeek-R1 (Guo et al., 2025), and Kimi k1.5 (Team et al., 2025). A notable feature of this type of LLMs is the phenomenon called Test-Time Scaling (TTS) (Zhang et al., 2025c), which generates long CoTs to scale performance. Reinforcement Learning with Verifiable Rewards (RLVR) (Mroueh, 2025) has been proven to be an effective technique for achieving TTS in the post-training process.

Recently, Reinforcement Learning (RL) methods have shown significantly better generalization performance in improving LLM reasoning capabilities compared to traditional Supervised Fine-Tuning (SFT) (Chu et al., 2025). SFT relies on high-quality, labeled data from human annotations or stronger model distillation, while RL relies primarily on the model's own exploration. Rollout-based reinforcement learning methods represented by Group Relative Policy Optimization (GRPO) (Shao et al., 2024) require the model to generate multiple trajectories for each training sample and learn based on the rewards of the generated trajectories, which can continuously expand the boundaries of LLMs' capabilities through the continuous RL process with diverse training samples.

However, existing rollout-based RL methods do not consider how well the model's current abilities match the difficulty of training samples. In human learning, people usually start with easy tasks and move to harder ones, an approach called Curriculum Learning (CL) (Wang et al., 2022; Soviany et al., 2022). Rollout-based RL methods have the model explore rollouts generated by the training samples, without considering if those samples are easy or hard. This does not help LLMs learn efficiently from samples with different levels of difficulty. Also, the model's skills change during RL training, so the difficulty of training samples can vary for the model at different stages. Because of this, pre-sorting training samples by fixed difficulty is not effective.

---

[1] `https://openai.com/index/gpt-5-system-card`

To address these limitations, we introduce a curriculum reinforcement learning framework called VCRL. It dynamically adjusts the difficulty of training samples based on the variance of group rewards. We find that the variance in rollout group rewards in RLVR partly reflects how hard a sample is for LLMs. With RLVR's current sparse reward system, samples that are too hard often get only 0 rewards, leading to low variance; this also happens with samples that are too easy. When samples are more uncertain, such as when half of the rollouts receive a reward of 1 and the other half receive 0, the model is at a key learning point for that sample. VCRL uses **Variance-based Dynamic Sampling** to select these samples for training, helping control the quality of the training batch. Group variance also gives a way to measure sample difficulty for the current state of the model. Therefore, VCRL uses **Replay Learning** with a memory bank to further boost training efficiency.

Our contributions are as follows:

- We introduce VCRL, a curriculum reinforcement learning framework that adjusts the difficulty of training samples based on the variance of group rewards. By focusing on samples with high reward variance, VCRL selects those most valuable for current model training.
- Building on group variance, we further introduce Replay Learning with a memory bank to control training stability and improve training efficiency. By updating and utilizing the memory bank, VCRL ensures high variance of samples in the training batch, thus achieving higher training value.
- We conduct extensive experiments on five benchmark datasets to justify VCRL's advantage on LLM's efficient Test-Time Scaling over some SOTA RL methods. Our results show consistent performance gains across different models, validating the effectiveness and robustness of our VCRL.

## 2 PRELIMINARIES

In this section, we review the current policy-based reinforcement learning methods in LLM, especially the rollout-based like GRPO and some variants.

### 2.1 PROXIMAL POLICY OPTIMIZATION (PPO)

PPO (Schulman et al., 2017) limits the update of the current policy to the proximal region of the old policy through the clipping mechanism. Specifically, give a dataset $\mathcal{D}$, $x$ is the query and $y$ is the response. For the policy model $\pi_\theta$ parameterized by $\theta$, the likelihood by the policy $\pi_\theta$ is given by $\pi_\theta(y|x) = \prod_{t=1}^{|y|} \pi_\theta(y_t|x, y_{<t})$, where $|y|$ is the number of tokens in $y$. In RLVR, there is a verifier $r$ that can score a given query-response pair $(x, y)$ and obtain a reward $r(x, y) \in [0, 1]$. PPO optimizes the following objective for policy optimization to update the actor in the proximal region of the old policy $\pi_{\theta_{old}}$:

$$\mathcal{J}_{\text{PPO}}(\theta) = \mathbb{E}_{x \sim \mathcal{D}, y \sim \pi_{\theta_{old}}(\cdot|x)} \left[ \frac{1}{|y|} \sum_{t=1}^{|y|} \min\left( r_t(\theta)\hat{A}_t, \text{clip}(r_t(\theta), 1-\epsilon, 1+\epsilon)\hat{A}_t \right) \right], \quad (1)$$

where the importance ratio of the token $y_t$ is given by $r_t(\theta) = \frac{\pi_\theta(y_t|x, y_{<t})}{\pi_{\theta_{old}}(y_t|x, y_{<t})}$, $\epsilon$ is the clipping range of the importance ratio, and the advantage $\hat{A}_t$ of $y_t$ is estimated using a value model estimated by Generalized Advantage Estimator (GAE) (Schulman et al., 2016).

PPO relies on the value model to evaluate the current state. Typically, the value model and the trained model have similar structures and parameters, resulting in significant computational and memory costs. Furthermore, the accuracy of the value model itself limits the effectiveness of the PPO algorithm, especially for long response and sparse reward in complex tasks for LLM.

### 2.2 GROUP RELATIVE POLICY OPTIMIZATION (GRPO) AND VARIANTS

GRPO (Shao et al., 2024) calculates the relative advantages of each response within a group of responses generated by LLM to the same query, eliminating the need to the value model. Specifically,

GRPO optimizes the following objective for policy optimization to update the actor within the group of responses (we omit the KL regularization term for brevity):

$$\mathcal{J}_{\text{GRPO}}(\theta) = \mathbb{E}_{x \sim \mathcal{D}, \{y_i\}_{i=1}^G \sim \pi_{\theta_{\text{old}}}(\cdot|x)}$$
$$\left[ \frac{1}{G} \sum_{i=1}^G \frac{1}{|y_i|} \sum_{t=1}^{|y_i|} \min \left( r_{i,t}(\theta) \hat{A}_{i,t}, \text{clip}(r_{i,t}(\theta), 1 - \epsilon, 1 + \epsilon) \hat{A}_{i,t} \right) \right], \quad (2)$$

where $G$ is the number of generated responses to the same query $x$, the importance ratio $r_{i,t}(\theta)$ and advantage $\hat{A}_{i,t}$ of token $y_{i,t}$ are given by

$$r_{i,t}(\theta) = \frac{\pi_\theta(y_{i,t}|x, y_{i,<t})}{\pi_{\theta_{\text{old}}}(y_{i,t}|x, y_{i,<t})}, \ \hat{A}_{i,t} = \frac{r(x, y_i) - \text{mean}\left(\{r(x, y_i)\}_{i=1}^G\right)}{\text{std}\left(\{r(x, y_i)\}_{i=1}^G\right)}. \quad (3)$$

Based on GRPO, Decoupled Clip and Dynamic sampling Policy Optimization (DAPO) (Yu et al., 2025) removes the KL divergence regularization and introduces the clip-higher and dynamic sampling with token-level loss, further improving the training stability and performance for LLMs. Specifically, DAPO optimizes the following objective for policy optimization:

$$\mathcal{J}_{\text{DAPO}}(\theta) = \mathbb{E}_{x \sim \mathcal{D}, \{y_i\}_{i=1}^G \sim \pi_{\theta_{\text{old}}}(\cdot|x)}$$
$$\left[ \frac{1}{\sum_{i=1}^G |y_i|} \sum_{i=1}^G \sum_{t=1}^{|y_i|} \min \left( r_{i,t}(\theta) \hat{A}_{i,t}, \text{clip}(r_{i,t}(\theta), 1 - \epsilon_{\text{low}}, 1 + \epsilon_{\text{high}}) \hat{A}_{i,t} \right) \right], \quad (4)$$
$$\text{s.t.} \ 0 < |\{y_i | \texttt{is\_equivalent}(y^*, y_i)\}| < G,$$

where $\epsilon_{\text{low}}$ and $\epsilon_{\text{high}}$ are the low and high clipping bound for the importance ratio respectively, and $y^*$ is the correct answer.

Based GRPO, Group Sequence Policy Optimization (GSPO) (Zheng et al., 2025) uses sequence-level importance ratio to replace the original token-level importance ratio to match the sentence-level reward in the generation task and optimization objective, thus achieving remarkable improvements. Specifically, GSPO optimizes the following objective for policy optimization:

$$\mathcal{J}_{\text{GSPO}}(\theta) = \mathbb{E}_{x \sim \mathcal{D}, \{y_i\}_{i=1}^G \sim \pi_{\theta_{\text{old}}}(\cdot|x)} \left[ \frac{1}{G} \sum_{i=1}^G \min \left( s_i(\theta) \hat{A}_i, \text{clip}(s_i(\theta), 1 - \epsilon, 1 + \epsilon) \hat{A}_i \right) \right], \quad (5)$$

where the group-based advantage estimation and importance ratio are given by

$$\hat{A}_i = \frac{r(x, y_i) - \text{mean}\left(\{r(x, y_i)\}_{i=1}^G\right)}{\text{std}\left(\{r(x, y_i)\}_{i=1}^G\right)}, \ s_i(\theta) = \left( \frac{\pi_\theta(y_i|x)}{\pi_{\theta_{\text{old}}}(y_i|x)} \right)^{\frac{1}{|y_i|}}. \quad (6)$$

# 3 VARIANCE-BASED CURRICULUM REINFORCEMENT LEARNING

In this section, we introduce Variance-based Curriculum Reinforcement Learning (VCRL), shown in Figure 1. First, we explain Variance-based Dynamic Sampling and how it helps identify the difficulty and value of training samples. Next, we combine Replay Learning with a memory bank to focus training on high-value samples, which improves RL training efficiency and stability.

## 3.1 VARIANCE-BASED DYNAMIC SAMPLING

As discussed above, existing rollout-based RL methods do not properly match model capabilities with sample difficulty during training. This problem mainly shows up in two ways:

1. **Dynamic Model Parameters**: During training, gradient backpropagation is performed using the objective function calculated from the training samples. This updates the model parameters to improve its performance on current samples. Model parameters keep changing, so the model may perform differently on the same samples at different stages of training.

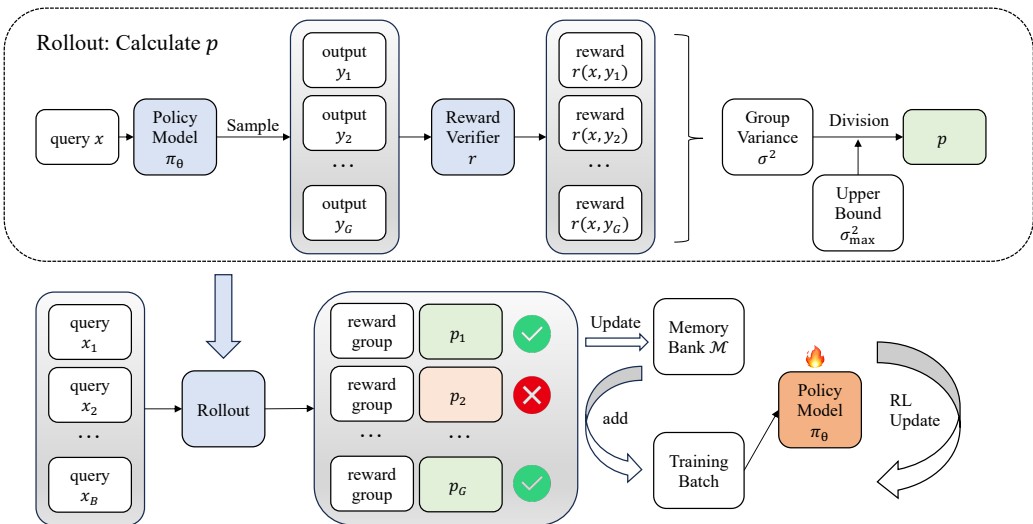

Figure 1: An illustration of the proposed VCRL method. For rollout-based RL training, VCRL first calculates our proposed $p$ for each query's rollout results and filters queries based on their $p$. VCRL then uses the existing memory bank $\mathcal{M}$ to update and add training samples. Finally, VCRL performs the standard RL update for this training batch.

2. **Unordered Sample Difficulty**: For most training datasets and algorithms, the difficulty of training samples is not considered. Some tasks, like search (Feng et al., 2025b; Hao et al., 2025) and tool use (Shen, 2024; Lu et al., 2025), are hard to define by difficulty. Also, sorting samples by difficulty requires a lot of data preprocessing. As a result, most datasets include samples that are not ordered by difficulty.

Dynamic model parameters and unordered sample difficulty make it too expensive and hard to use ordered training samples based on predefined difficulty. Samples that are hard for the model early in training often become easier later. So, the indicator of training sample difficulty must be adjusted dynamically as the model changes.

Multiple rollouts for the same query can help measure how hard a training sample is for the current model. Formally, for a query $x$, if it is too easy for model $\pi_\theta$, then $\mathbb{E}_{y \sim \pi_\theta(\cdot|x)} [r(x, y)] \approx 1$. If $x$ is too hard, then $\mathbb{E}_{y \sim \pi_\theta(\cdot|x)} [r(x, y)] \approx 0$. Both easy and hard samples have low group reward variance. So, we can use the variance of group rewards to pick samples that are better suited for the current model. Samples with higher variance are neither too easy nor too hard, meaning the difference between the probabilities of positive and negative outcomes is small.

In RLVR, for the binary reward distribution, the group variance for the query $x$ is given by

$$\text{Var}_{y \sim \pi_\theta(\cdot|x)}(r(x, y)) = \mathbb{E}_{y \sim \pi_\theta(\cdot|x)} \left[ (r(x, y) - \mathbb{E}_{y \sim \pi_\theta(\cdot|x)} [r(x, y)])^2 \right]. \tag{7}$$

If there are $k$ rollouts with a reward of 1, the unbiased estimator of the group variance can be written as

$$\sigma^2 = \frac{1}{G-1} \sum_{i=1}^{G} \left[ r(x, y_i) - \frac{1}{G} \sum_{i=1}^{G} r(x, y_i) \right]^2$$

$$= \frac{1}{G-1} \sum_{i=1}^{G} \left[ r(x, y_i) - \frac{k}{G} \right]^2$$

$$= \frac{k(G-k)}{G(G-1)}. \tag{8}$$

When $k = \lfloor \frac{G}{2} \rfloor$, the estimator achieves the maximum value:

$$\sigma_{\max}^2 = \begin{cases} \frac{G}{4(G-1)}, & G \text{ is even,} \\ \frac{G+1}{4G}, & G \text{ is odd.} \end{cases} \tag{9}$$

Obviously, the group variance cannot exceed $\sigma_{\max}^2$ in any case, so we can use the normalized group variance $p = \frac{\sigma^2}{\sigma_{\max}^2}$ to measure the value of the current query $x$ for the model $\pi_\theta$. Training with samples that have high $p$ helps the model learn areas where it is less skilled, which improves the model more effectively than using unordered samples. See Appendix Section B for more discussion.

### 3.2 REPLAY LEARNING

Based on the normalization value $p$ discussed above, we can dynamically sample queries during training using threshold rules. This helps ensure that each training sample has high value for the model. For unordered training datasets, each sampled query can only obtain its $p$ value after a long rollout, so we use variance-based dynamic sampling. Calculating $p$ for each training sample requires significant computational resources and time, which can be expensive if used only for sampling.

To address this, we propose building a high-value memory bank using $p$ and maintaining it with a momentum update method. This lets us apply curriculum learning with data replay based on group variance, as shown in Algorithm 1. Specifically, each time we sample from the training set $\mathcal{D}$, we get a query batch $\{x_j\}_{j=1}^B$, where $B$ is the batch size. First, we get the corresponding response set $\{y_{j,i}\}_{i=1}^G$ and reward set $\{r(x_j, y_{j,i})\}_{i=1}^G$, then calculate $p_j$ for each query $x_j$. If $p_j \geq \kappa$, where $\kappa \in [0, 1]$ is a predefined threshold, we keep the query $x_j$. Otherwise, we remove it from the batch and perform variance-based dynamic sampling.

Suppose $M$ queries are removed from a batch of $B$. To keep the batch size unchanged, we replace the missing $M$ queries by sampling queries from the memory bank $\mathcal{M}$. The memory bank $\mathcal{M}$ is implemented as a priority queue, where each entry is a query $x_j$, and the priority $P(x_j)$ is updated based on momentum and the number of steps since it was last accessed, $\beta(x_j)$:

$$P(x_j) \leftarrow \alpha P(x_j) + (1 - \alpha)\beta(x_j), \tag{10}$$

where $\alpha$ is the momentum constant and the $P(x_j)$ is initialized using $p_j$.

The proposed VCRL based on GRPO optimizes the following objective for policy optimization:

$$\mathcal{J}_{\text{VCRL}}(\theta) = \mathbb{E}_{x \sim \mathcal{D} \cup \mathcal{M}, \{y_i\}_{i=1}^G \sim \pi_{\theta_{\text{old}}}(\cdot|x)}$$

$$\left[ \frac{1}{G} \sum_{i=1}^G \frac{\mathbb{I}\left(p_i = \frac{\sigma_i^2}{\sigma_{\max}^2} \geq \kappa\right)}{|y_i|} \sum_{t=1}^{|y_i|} \min\left(r_{i,t}(\theta)\hat{A}_{i,t}, \text{clip}(r_{i,t}(\theta), 1 - \epsilon, 1 + \epsilon)\hat{A}_{i,t}\right) \right], \tag{11}$$

where the calculation of $p_i$ and the memory bank $\mathcal{M}$ mechanism are as described above, and $\mathbb{I}(\cdot)$ is the indicator function. See Appendix Section C for a comparison of theoretical perspecitves on GRPO and VCRL.

## 4 EXPERIMENTS

### 4.1 EXPERIMENTAL SETUP

**Benchmarks.** In this work, we focus specifically on mathematical reasoning tasks to evaluate our VCRL algorithm. For mathematical reasoning tasks, we use AIME-2024[2], AIME-2025[3], MATH500 (Lightman et al., 2024), OlympiadBench (He et al., 2024), and AMC23[4]. Among them, AIME-2024 and AIME-2025 are used as high-difficulty benchmarks to effectively evaluate the performance of VCRL and other baseline RL methods in multiple difficulty levels.

---

[2] https://huggingface.co/datasets/Maxwell-Jia/AIME_2024
[3] https://huggingface.co/datasets/yentinglin/aime_2025
[4] https://huggingface.co/datasets/AI-MO/aimo-validation-amc

---

**Algorithm 1:** VCRL: Variance-based Curriculum Reinforcement Learning

---

**Require:** Training Set $\mathcal{D}$, Reward Verifier $r$, $p$-threshold $\kappa$, Policy Model $\pi_\theta$, Momentum Constant $\alpha$, Training Batch Size $B$, Rollout Group Size $G$

1: Initialize $\mathcal{M} \leftarrow$ PriorityQueue()
2: **while** Training **do**
3:     Sample $\{x_j\}_{j=1}^B \sim \mathcal{D}$, $M \leftarrow 0$
4:     **for** $j = 1$ to $B$ **do**
5:         Sample $\{y_{j,i}\}_{i=1}^G \sim \pi_\theta(\cdot|x_j)$
6:         Calculate Reward $\{r(x_j, y_{i,j})\}_{i=1}^G$
7:         Calculate $p_j$ for $x_j$
8:         **if** $p_j < \kappa$ **then**
9:             Remove $x_j$ from Training Batch
10:           $M \leftarrow M + 1$
11:         **end if**
12:     **end for**
13:     Pop $M$ queries from $\mathcal{M}$ and add them to the Training Batch
14:     **for** $x \in \mathcal{M}$ **do**
15:         $\beta(x) \leftarrow \beta(x) + 1$
16:         $P(x) \leftarrow \alpha P(x) + (1 - \alpha)\beta(x)$
17:     **end for**
18:     Apply RL update using the Augmented Training Batch $\mathcal{B}$
19:     **for** $x \in \mathcal{B}$ **do**
20:         Calculate $p$ for $x$
21:         **if** $p \geq \kappa$ **then**
22:             Push $x$ into $\mathcal{M}$ with priority $P(x) = p$ and $\beta(x) = 0$
23:         **end if**
24:     **end for**
25: **end while**

---

**Implementation Details.** For training dataset, we use DAPO-Math-17K[5] to improve training stability, which consists of 17K prompts, each paired with an interger as the answer. We implement VCRL and conduct all experiments based on the verl (Sheng et al., 2025) framework. For hyperparameters, we utilize the AdamW (Loshchilov & Hutter, 2019) optimizer with a constant learning rate of $1 \times 10^{-6}$. For rollout, the prompt batch size is $B = 128$ and we sample $G = 16$ responses for each prompt. For training, we train 500 steps to ensure convergence. The maximum number of tokens for generation is set to 4,096 tokens. For evaluation on benchmarks, we repeat the evaluation set for 16 times and report avg@16 for the stability of the results. The inference hyperparameters of evaluation are set to temperature 0.6 and top-p 0.95. For VCRL, we set the variance threshold $\kappa$ to 0.3 in first 20 steps and 0.8 in remaining steps, and the momentum constant $\alpha$ is set to 0.9. We implement VCRL based on GRPO's RL update. For memory bank, we allow up to 2 replays for the same sample to ensure the diversity of training sample. We conduct all experiments on a server with $8 \times$NVIDIA H20-3e GPUs and an Intel® Xeon® Platinum 8575C CPU.

**Baselines and Models.** We mainly use GRPO (Shao et al., 2024), DAPO (Yu et al., 2025) and GSPO (Zheng et al., 2025) as the baselines for our VCRL comparison. For Clip-Higher mechanism in DAPO, we set the clipping parameter $\epsilon_{\text{low}}$ to 0.2 and $\epsilon_{\text{high}}$ to 0.28, which is aligned with the DAPO setting in the original paper. For GSPO, we set the clipping parameter $\epsilon$ to 0.0003. For models, we use the Qwen3 (Yang et al., 2025a) series models for training, including *Qwen3-4B-Base* and *Qwen3-8B-Base*.

## 4.2 MAIN RESULTS

We conduct a comprehensive evaluation of our proposed method, VCRL, against several strong LLM RL baselines on a diverse suite of mathematical reasoning benchmarks. As detailed in Table 1, the experiments are performed on two models, *Qwen3-4B-Base* and *Qwen3-8B-Base*, to assess the

---

[5]https://huggingface.co/datasets/BytedTsinghua-SIA/DAPO-Math-17k

Table 1: Main performance comparison of VCRL against other RL baselines on Qwen3 models.

| Method | AIME-2024 | AIME-2025 | MATH500 | OlympiadBench | AMC23 | Avg. |
|---|---|---|---|---|---|---|
| *Starting from Qwen3-4B-Base* | | | | | | |
| Base Model | 9.58 | 4.79 | 56.69 | 27.27 | 35.09 | 26.68 |
| + GRPO | 15.63 | 12.92 | 80.78 | 45.39 | 54.07 | 41.76 |
| + DAPO | 14.79 | 12.29 | 79.86 | 44.23 | 51.81 | 40.60 |
| + GSPO | 14.58 | 10.42 | 79.90 | 44.38 | 51.13 | 40.08 |
| + VCRL | **23.96** | **22.71** | **86.48** | **53.24** | **60.77** | **49.43** |
| *Starting from Qwen3-8B-Base* | | | | | | |
| Base Model | 10.83 | 10.00 | 68.75 | 34.10 | 41.11 | 32.96 |
| + GRPO | 23.13 | 21.88 | 86.94 | 54.02 | 65.29 | 50.25 |
| + DAPO | 22.08 | 20.42 | 87.14 | 53.52 | 64.01 | 49.43 |
| + GSPO | 27.29 | 22.92 | 89.23 | 56.75 | 69.28 | 53.09 |
| + VCRL | **34.38** | **27.08** | **91.99** | **60.21** | **75.15** | **57.76** |

scalability and generalizability of our method. The results unequivocally demonstrate the superiority of VCRL. Across all five benchmarks and on both model sizes, VCRL consistently achieves state-of-the-art performance, outperforming all baseline methods, including GRPO, DAPO, and GSPO. This consistent dominance, indicated by the bolded scores, highlights the robustness and effectiveness of our proposed methodology.

A deeper analysis reveals the substantial performance gains enabled by VCRL. For instance, on the *Qwen3-8B-Base* model, VCRL achieves an average score of 57.76, a significant margin of over 4.67 points above the strongest baseline, GSPO (53.09), and a remarkable 24.8 points improvement over the base model. This trend holds for the *Qwen3-4B-Base* model, where VCRL elevates the average performance from 26.68 (Base Model) to 49.43, far surpassing the gains from other RL techniques. Notably, the performance leap is particularly pronounced on highly challenging, competition-level datasets such as AIME-2024 and AIME-2025, suggesting that VCRL is exceptionally proficient at unlocking the complex, multi-step reasoning capabilities essential for advanced mathematical problem-solving. These empirical findings strongly validate VCRL as a superior alignment strategy for enhancing the mathematical reasoning prowess of LLMs.

## 4.3 PERFORMANCE TREND

During RL training, the LLM starts with low ability and steadily improves, showing an upward trend on benchmark tests. To illustrate how VCRL compares to baseline methods during training, we show how model performance changes with training steps on each benchmark, as seen in Figure 2 for *Qwen3-4B-Base* and Figure 3 for *Qwen3-8B-Base*.

For performance trend, the results clearly demonstrate that VCRL consistently and significantly outperforms all other baseline methods across all benchmarks. In terms of the speed of performance improvement, VCRL also has a considerable advantage. In the first 100 training steps, VCRL's performance increases quickly, with its curve staying above the other methods. This is likely due to VCRL's control of high-$p$ training samples in the early stages of model training, which improves training efficiency. Later in training, the performance of all methods generally converges, but VCRL still achieves significantly better final results than the RL baselines. This demonstrates VCRL's strong competitiveness. More training dynamics are in Appendix Section A.

## 4.4 ABLATION STUDY

To verify the effectiveness of the two core components of our proposed VCRL, we conduct the ablation study, as shown in Table 2. Starting from the *Qwen3-4B-Base*, our Naive GRPO baseline improves the average score from 26.68 to 41.76. The integration of Variance-based Dynamic Sampling further pushes this score to 44.73. Finally, the inclusion of Replay Learning achieves the best performance of 49.43, showing the largest marginal gain. This consistent trend on the larger *Qwen3-8B-Base* model robustly validates the positive impact of each component within our VCRL framework.

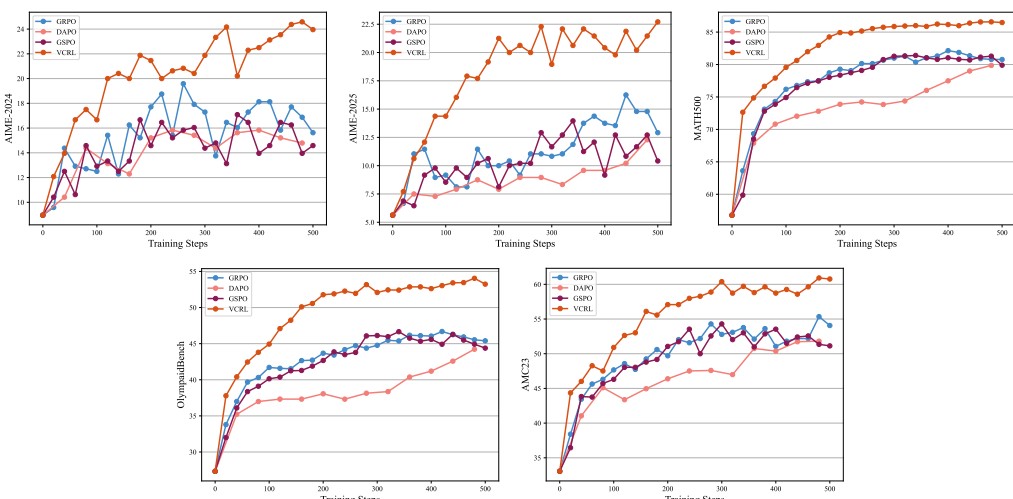

Figure 2: The performance curve of *Qwen3-4B-Base* on the five benchmarks using various RL methods over training steps.

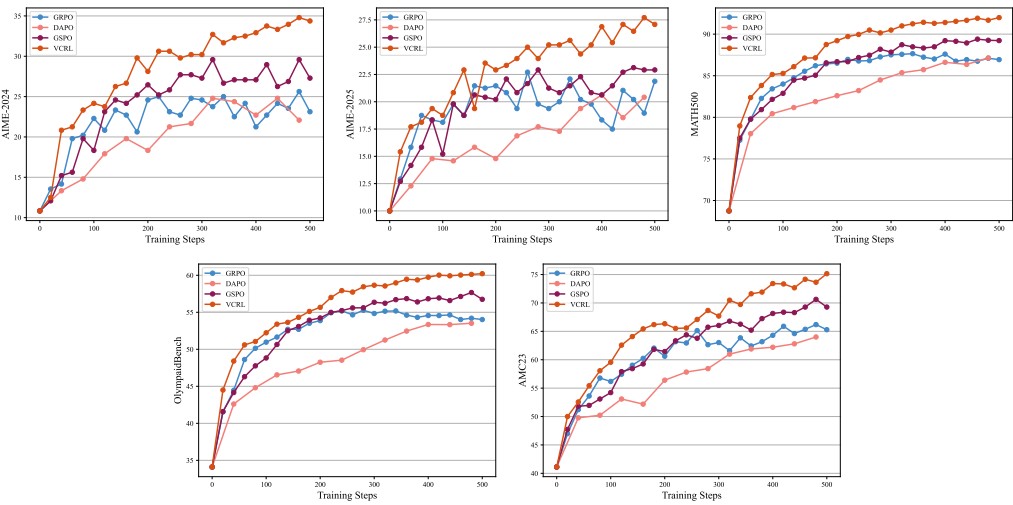

Figure 3: The performance curve of *Qwen3-8B-Base* on the five benchmarks using various RL methods over training steps.

## 5 RELATED WORK

Recent work on using RL methods with LLMs has greatly improved their ability to handle complex tasks. DeepSeek-R1 (Guo et al., 2025) introduces a zero RL training framework, which directly trains the base LLM using a simple rule-based reward model. Many RL methods have built on this idea to further boost LLM performance.

Some approaches use novel RL mechanisms to make training more efficient and stable. DAPO (Yu et al., 2025) analyzes GRPO's training and applies four main techniques to improve RL efficiency. Dr. GRPO (Liu et al., 2025b) removes the output length and standard deviation terms from GRPO's relative advantage, which increases token efficiency without hurting reasoning performance. SimpleRL-Zoo (Zeng et al., 2025) runs experiments on different base models and sizes to map out behavioral patterns and suggest future improvements. LUFFY (Yan et al., 2025) enhances RLVR with off-policy reasoning traces, helping to balance imitation and exploration by combining off-policy demonstrations with on-policy rollouts. VAPO (Yue et al., 2025) introduces the first value-model-based RL training framework built on PPO, with seven new techniques to improve

Table 2: Ablation study of the key components of our proposed method VCRL. Starting from a Naive GRPO baseline, we incrementally add **Variance-based Dynamic Sampling** and **Replay Learning**. The results on both *Qwen3-4B-Base* and *Qwen3-8B-Base* models show that each component contributes positively to the final performance, validating their effectiveness.

| Model | Avg. |
|---|---|
| *Qwen3-4B-Base* | 26.68 |
| w/ Naive GRPO | 41.76 |
| w/ Variance-based Dynamic Sampling | 44.73 |
| w/ Replay Learning | **49.43** |
| *Qwen3-8B-Base* | 32.96 |
| w/ Naive GRPO | 50.25 |
| w/ Variance-based Dynamic Sampling | 52.67 |
| w/ Replay Learning | **57.76** |

training stability and performance. Yeo et al. (2025) investigates how RL helps models create longer reasoning chains, showing which factors matter most for extended CoT reasoning. PVPO (Feng et al., 2025c) presents an efficient reinforcement learning method enhanced by an advantage reference anchor and data pre-sampling.

Other work explores curriculum learning in LLM training for better results. Hammoud et al. (2025) improve GRPO with a reward function that balances task correctness (via verifier feedback), length efficiency, and formatting (using structural tags), leading to higher accuracy and better token efficiency. Feng et al. (2025a) propose a self-adaptive curriculum that picks fine-tuning examples based on difficulty scores predicted by pre-trained models. Shen et al. (2025) introduce TTI (Test-Time Interaction), an online RL method that adapts rollout lengths using a curriculum approach. Parashar et al. (2025) provide convergence guarantees for easy-to-hard training within an approximate policy iteration framework. RAGEN (Wang et al., 2025c) introduces uncertainty-based filtering to maintain high training efficiency based on active learning (Settles, 2009). PODS (Xu et al., 2025b) generates numerous rollouts in parallel but updating only on informative subset. Curr-ReFT (Deng et al., 2025) explores the Out-of-Distribution generalization on small-scale Vision Language Models based on the curriculum learning framework. Xi et al. (2024) introduce a novel method that employs only outcome supervision to achieve the benefits of process supervision for large language models with a step-wise curriculum. More recently, several studies have focused on difficulty-based filtering and distribution priors. ADCL (Zhang et al., 2025a) employs adaptive difficulty curriculum via periodic data reordering, while other approaches (Chen et al., 2025; Wang et al., 2025b) rely on self-evolving schedules or distribution-level priors which often require preset difficulty hierarchies. Regarding data efficiency, Bae et al. (2025) propose ODF with a pass-based selection strategy similar to DAPO, and Tzannetos et al. (2023) explore the theoretical foundations of proximal curriculum. Distinct from these methods, our VCRL operates dynamically by filtering based on group variance within the current batch and utilizes replay learning to enhance efficiency, without assuming future data streams or requiring strict difficulty distribution priors.

RLVR (Mroueh, 2025) is a promising method for boosting reasoning in LLMs, especially in areas like math and programming (Jiang et al., 2025b). Gandhi et al. (2025) show that reasoning behaviors—not just correct answers—drive RL performance gains. Li et al. (2025) find that the structure of long chains of thought is key for learning, while the details of each reasoning step matter less. Vassoyan et al. (2025) identify critical tokens in CoTs, which are decision points where models often make mistakes, and suggest increasing exploration around these tokens by changing the KL penalty. Lin et al. (2024) also find tokens that lead to errors and show that changing them can shift model behavior.

## 6 CONCLUSION

In this paper, we propose VCRL, a curriculum reinforcement learning framework that dynamically controls the difficulty of training samples based on the variance of group rewards. By introducing Dynamic Variance Sampling, VCRL can filter out samples in the training batch that are moderately difficult for the current training model and remove samples that are too difficult or too easy, thereby

improving training efficiency. By introducing Replay Learning, VCRL uses a memory bank to maintain the high-$p$ samples in the training batch, further improving training stability. By carefully controlling the difficulty of training samples, VCRL achieves state-of-the-art results on five math benchmarks compared to LLM RL baselines. Further analysis of training dynamics and ablation study also confirm VCRL's effectiveness.

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

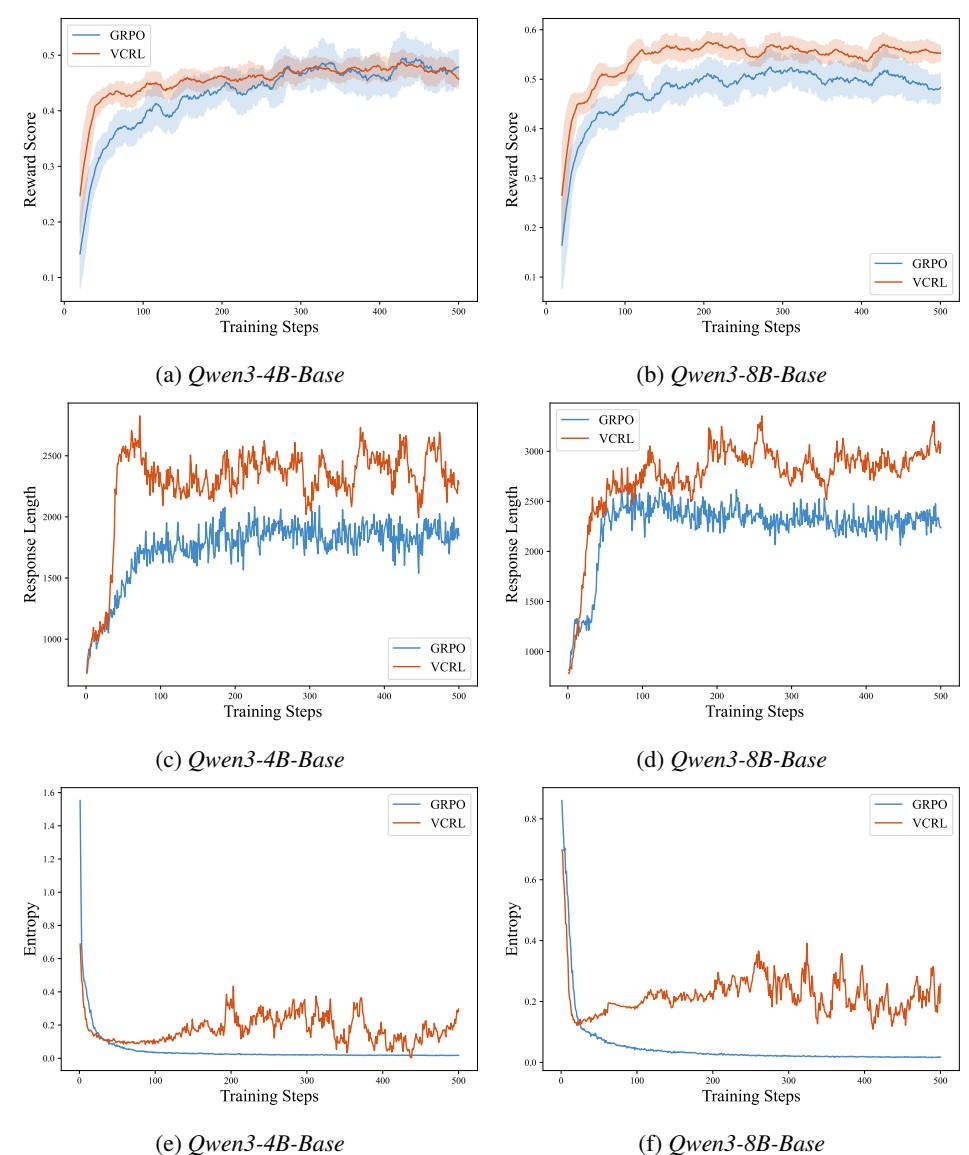

(a) *Qwen3-4B-Base*

(b) *Qwen3-8B-Base*

(c) *Qwen3-4B-Base*

(d) *Qwen3-8B-Base*

(e) *Qwen3-4B-Base*

(f) *Qwen3-8B-Base*

Figure 4: The metric curves of reward score, response length, and entropy of VCRL over GRPO based on *Qwen3-4B-Base* and *Qwen3-8B-Base*, which show the dynamics of RL training and serve as essential monitoring indicators to identify potential issues.

Zikang Zhang, Wangjie You, Tianci Wu, Xinrui Wang, Juntao Li, and Min Zhang. A survey of generative information extraction. In Owen Rambow, Leo Wanner, Marianna Apidianaki, Hend Al-Khalifa, Barbara Di Eugenio, and Steven Schockaert (eds.), *Proceedings of the 31st International Conference on Computational Linguistics, COLING 2025, Abu Dhabi, UAE, January 19-24, 2025*, pp. 4840–4870. Association for Computational Linguistics, 2025d. URL https://aclanthology.org/2025.coling-main.324/.

Chujie Zheng, Shixuan Liu, Mingze Li, Xiong-Hui Chen, Bowen Yu, Chang Gao, Kai Dang, Yuqiong Liu, Rui Men, An Yang, et al. Group sequence policy optimization. *arXiv preprint arXiv:2507.18071*, 2025.

## A  TRAINING DYNAMICS

Compared to GRPO, VCRL introduces two main techniques to improve training efficiency. To further understand their effects, we show the training dynamics shown in Figure 4, including reward score, response length, and entropy. For the reward score curve, in order to simultaneously measure their stability in training dynamics, we use moving average and rolling standard deviation with a window size of 20 for visualization.

- **Reward Score** during training is closely linked to training stability and performance, as shown in Figure 4a and Figure 4b. For both VCRL and GRPO, the reward score rises quickly in the early stages and then slowly improves. For *Qwen3-4B-Base*, before about 270 training steps, VCRL's reward score is much higher than GRPO's. For *Qwen3-8B-Base*, the reward score of VCRl is significantly higher than that of GRPO throughput the training process. Once the reward score stabilizes, VCRL shows much smaller fluctuations than GRPO, as seen in the shaded areas. This highlights VCRL's advantage in training stability.

- **Response Length** relates to how much the model can explore, as shown in Figure 4c and Figure 4d. Longer responses help the model develop more complex reasoning during training and boost performance. In the first 100 steps, VCRL and GRPO both show a rapid increase in response length, then level off and fluctuate. VCRL's response length grows much faster early on, especially in first 50 steps, due to the training of high-$p$ samples. After stabilizing, VCRL maintains noticeably longer responses, giving the model more room to explore and optimize its performance.

- **Entropy** shows how uncertain the model is in its generation ability, as seen in Figure 4e and Figure 4f. For efficient training, entropy should stay at a reasonable level. If entropy is too low, the model becomes too deterministic and loses its ability to explore. For GRPO, entropy quickly drops below 0.1 within 50 steps and stays very low. In contrast, VCRL keeps entropy at a reasonable level throughout training, which encourages the model to keep exploring.

## B  VARIANCE AS A DIFFICULTY METRIC

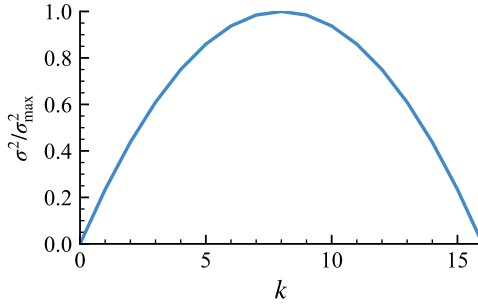

Figure 5: The curve showing how $p = \frac{\sigma^2}{\sigma^2_{\max}}$ changes with the number of successful rollouts $k$ based on group size $G = 16$.

Compared with generation entropy, it is more reasonable to use group variance to measure the difficulty of the current sample for the current training model in VCRL or GRPO. Group reward variance is grounded in its unique ability to identify samples at the cusp of the model's current capabilities.

For a binary reward system (correct/incorrect), variance exhibits a non-monotonic, U-shaped relationship with sample difficulty, as shown in Figure 5. Low variance occurs at two extremes. If a sample is too easy, the model consistently succeeds (e.g., all 16 rollouts get a reward of 1), leading to near-zero variance. If a sample is too hard, the model consistently fails (all rewards are 0), also leading to near-zero variance. Peaks when the model's success rate is approximately 50% (e.g., 8

rollouts succeed and 8 fail). This indicates maximum uncertainty and signifies that the sample is at the precise frontier of the model's ability.

While related to uncertainty, policy generation entropy measures the diversity of the model's actions (tokens). High entropy could mean the model is exploring, but it does not directly map to task-level success. A model could be highly uncertain (high entropy) while generating non-sensical responses that all lead to a reward of 0. Variance, on the other hand, is directly related to the final outcome of the task (the reward), making it a more direct measure of the difficulty relevant to learning. By using a single indicator of group variance, it is possible to filter samples with high uncertainty results, while this task is difficult to accomplish based on the generation entropy.

## C  POLICY GRADIENT REDUCTION

According to Equation 2 and Equation 11, we give the following theorem:

**Theorem 1.** *For policy gradient algorithm GRPO and VCRL, from the policy gradient norm perspective, the training of VCRL is more stable than that of GRPO in the expectation, that is,* $\mathbb{E}_{VCRL}\left[\|\nabla_\theta \log \pi_\theta\|\right] \le \mathbb{E}_{GRPO}\left[\|\nabla_\theta \log \pi_\theta\|\right]$.

*Proof.* We first give the gradient form of the GRPO objective function (clipping is omitted for brevity) with Policy Gradient Theorem (Sutton & Barto, 1998):

$$\nabla_\theta \mathcal{J}_{\text{GRPO}}(\theta) = \nabla_\theta \mathbb{E}_{x \sim \mathcal{D}, \{y_i\}_{i=1}^G \sim \pi_{\theta_{\text{old}}}(\cdot|x)} \left[ \frac{1}{G} \sum_{i=1}^G \frac{1}{|y_i|} \sum_{t=1}^{|y_i|} r_{i,t}(\theta) \hat{A}_{i,t} \right]$$

$$= \mathbb{E}_{x \sim \mathcal{D}, \{y_i\}_{i=1}^G \sim \pi_{\theta_{\text{old}}}(\cdot|x)} \left[ \frac{1}{G} \sum_{i=1}^G \frac{1}{|y_i|} \sum_{t=1}^{|y_i|} r_{i,t}(\theta) \hat{A}_{i,t} \nabla_\theta \log \pi_\theta(y_{i,t}|x, y_{i,<t}) \right],$$

(12)

where $r_{i,t}(\theta) = \frac{\pi_\theta(y_{i,t}|x, y_{i,<t})}{\pi_{\theta_{\text{old}}}(y_{i,t}|x, y_{i,<t})}$ is the importance sampling ratio.

We can also derive the gradient of the VCRL objective as follows:

$$\nabla_\theta \mathcal{J}_{\text{VCRL}}(\theta) = \nabla_\theta \mathbb{E}_{x \sim \mathcal{D} \cup \mathcal{M}, \{y_i\}_{i=1}^G \sim \pi_{\theta_{\text{old}}}(\cdot|x)} \left[ \frac{1}{G} \sum_{i=1}^G \frac{\mathbb{I}\left(p_i = \frac{\sigma_i^2}{\sigma_{\max}^2} \ge \kappa\right)}{|y_i|} \sum_{t=1}^{|y_i|} r_{i,t}(\theta) \hat{A}_{i,t} \right]$$

$$= \mathbb{E}_{x \sim \mathcal{D} \cup \mathcal{M}, \{y_i\}_{i=1}^G \sim \pi_{\theta_{\text{old}}}(\cdot|x)}$$

$$\left[ \frac{1}{G} \sum_{i=1}^G \frac{\mathbb{I}\left(p_i = \frac{\sigma_i^2}{\sigma_{\max}^2} \ge \kappa\right)}{|y_i|} \sum_{t=1}^{|y_i|} r_{i,t}(\theta) \hat{A}_{i,t} \nabla_\theta \log \pi_\theta(y_{i,t}|x, y_{i,<t}) \right].$$

(13)

To align the gradients of the two, we use importance sampling to rewrite the gradient of VCRL to remove the term of memory bank $\mathcal{M}$:

$$\nabla_\theta \mathcal{J}_{\text{VCRL}}(\theta) = \mathbb{E}_{x \sim \mathcal{D}, \{y_i\}_{i=1}^G \sim \pi_{\theta_{\text{old}}}(\cdot|x)}$$

$$\left[ \frac{1}{G} \sum_{i=1}^G \frac{1}{|y_i|} \sum_{t=1}^{|y_i|} r_{i,t}(\theta) \hat{A}_{i,t} \frac{\color{blue}{\mathbb{P}(x \in \mathcal{D} \cup \mathcal{M})}}{\color{blue}{\mathbb{P}(x \in \mathcal{D})}} \mathbb{I}(p_i \ge \kappa) \nabla_\theta \log \pi_\theta(y_{i,t}|x, y_{i,<t}) \right].$$

(14)

Note that the blue part in the Equation 14 is the key to affecting the contribution of the policy gradient term to the overall gradient.

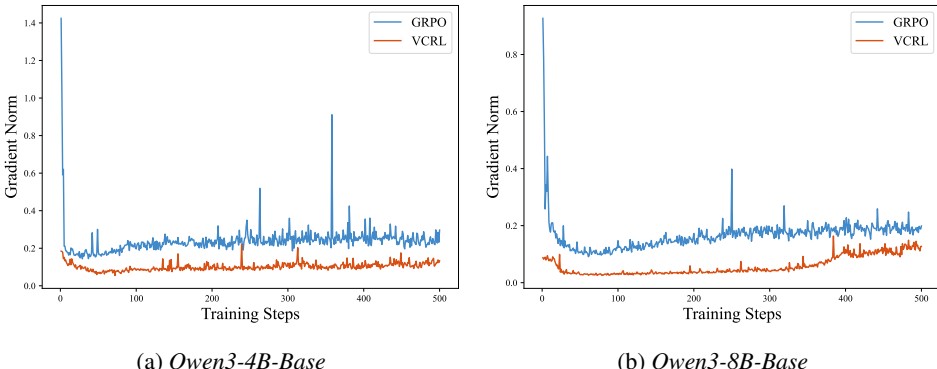

(a) *Qwen3-4B-Base*  (b) *Qwen3-8B-Base*

Figure 6: The training dynamics of objective gradient norm $\|\nabla_\theta \mathcal{J}(\theta)\|$ of VCRL over GRPO based on *Qwen3-4B-Base* and *Qwen3-8B-Base*.

We simplify the policy gradient terms in Equation 12 and Equation 14 into the following form for comparison:

$$\mathbb{E}_{\text{GRPO}}\left[\|\nabla_\theta \log \pi_\theta\|\right] = \mathbb{E}_{x \sim \mathcal{D}, \{y_i\}_{i=1}^G \sim \pi_{\theta_{\text{old}}}(\cdot|x)}\left[\|\nabla_\theta \log \pi_\theta(y_{i,t}|x, y_{i,<t})\|\right], \tag{15}$$

$$\mathbb{E}_{\text{VCRL}}\left[\|\nabla_\theta \log \pi_\theta\|\right] = \mathbb{E}_{x \sim \mathcal{D}, \{y_i\}_{i=1}^G \sim \pi_{\theta_{\text{old}}}(\cdot|x)}$$
$$\left[\left\|\frac{\mathbb{P}(x \in \mathcal{D} \cup \mathcal{M})}{\mathbb{P}(x \in \mathcal{D})}\mathbb{I}(p_i \geq \kappa)\nabla_\theta \log \pi_\theta(y_{i,t}|x, y_{i,<t})\right\|\right]. \tag{16}$$

For the training sample $x$, the sampling of the event $x \in \mathcal{D}$ is uniform, so $\mathbb{P}(x \in \mathcal{D}) = \frac{1}{|\mathcal{D}|}$. And according to the nature of sampling probability, we can get $\mathbb{P}(x \in \mathcal{D} \cup \mathcal{M}) \leq \mathbb{P}(x \in \mathcal{D})$. Based on the value range of the indicator function, we can also get $\mathbb{I}(p_i \geq \kappa) \leq 1$. Using the homogeneity of the norm and above results:

$$\left\|\frac{\mathbb{P}(x \in \mathcal{D} \cup \mathcal{M})}{\mathbb{P}(x \in \mathcal{D})}\mathbb{I}(p_i \geq \kappa)\nabla_\theta \log \pi_\theta(y_{i,t}|x, y_{i,<t})\right\|$$
$$= \frac{\mathbb{P}(x \in \mathcal{D} \cup \mathcal{M})}{\mathbb{P}(x \in \mathcal{D})}\mathbb{I}(p_i \geq \kappa)\|\nabla_\theta \log \pi_\theta(y_{i,t}|x, y_{i,<t})\|$$
$$\leq \mathbb{I}(p_i \geq \kappa)\|\nabla_\theta \log \pi_\theta(y_{i,t}|x, y_{i,<t})\|$$
$$\leq \|\nabla_\theta \log \pi_\theta(y_{i,t}|x, y_{i,<t})\|,$$

which completes the proof. $\qquad\square$

Theorem 1 provides a theoretical guarantee for the training stability of VCRL compared to GRPO. To further illustrate the training stability of VCRL from the perspective of gradient norm, we show the training dynamics as shown in the Figure 6.

Figure 6 provides an empirical validation of our proposed VCRL's stability by visualizing the norm of the objective function's gradient, $\|\nabla_\theta \mathcal{J}(\theta)\|$, over the training steps. We compare VCRL against the GRPO baseline on two model scales: *Qwen3-4B-Base* and *Qwen3-8B-Base*. The empirical results unequivocally demonstrate the superiority of VCRL in maintaining a well-behaved optimization trajectory. Specifically, VCRL's gradient norm remains consistently confined to a lower and narrower band, indicating that the policy updates are more measured and stable. Furthermore, the VCRL curve is notably smoother, with significantly fewer and less pronounced transient spikes compared to the GRPO baseline. The frequent, high-magnitude oscillations observed in GRPO's gradient norm are indicative of a more challenging optimization landscape, which can lead to inefficient and unstable training. We posit that the demonstrably smaller and more stable gradient norm engendered by VCRL is an important contributor to its enhanced training efficiency and robust performance.

Table 3: Performance comparison on mathematical reasoning benchmarks using Qwen2.5-7B.

| Method | AIME-2024 | AIME-2025 | MATH500 | AMC23 | Avg. |
|---|---|---|---|---|---|
| Base Model | 16.67 | 16.67 | 69.40 | 32.50 | 33.81 |
| + GRPO | 26.67 | 16.67 | 72.40 | 45.00 | 40.19 |
| + ADCL | 33.33 | 30.00 | 76.20 | 55.00 | 48.63 |
| + ADCL & EGSR | **36.67** | **33.33** | **81.80** | 55.00 | 51.70 |
| + VCRL (Ours) | 35.83 | 32.71 | 79.73 | **61.22** | **52.37** |

## D  BASELINE COMPARISON

To further demonstrate the competitiveness of our proposed method VCRL, we provide a supplementary comparison with recent curriculum reinforcement learning methods, specifically focusing on ADCL (Zhang et al., 2025a). ADCL is a difficulty-based curriculum method that employs periodic data reordering.

To ensure a fair comparison, we align our experimental settings strictly with the configurations reported in ADCL. We utilize the Qwen2.5-7B model as the base policy model. During training, we sample 8 rollouts for each query, with a global batch size of 1,024, a fixed learning rate of $1 \times 10^{-6}$, a maximum response length of 4,096, and a temperature parameter of 0.7. We evaluated performance across our four challenging mathematical benchmarks: AIME-2024, AIME-2025, MATH500, and AMC23.

As shown in Table 3, VCRL achieves the highest average performance (52.37), surpassing both the standard ADCL (48.63) and its enhanced version ADCL & EGSR (51.70). Notably, VCRL demonstrates a significant advantage on the AMC23 benchmark, outperforming ADCL by over 6 points. Beyond numerical improvements, VCRL offers distinct methodological advantages. ADCL relies on difficulty-based reordering of the training data (e.g., reordering every 100 training steps) and operates under the assumption of a known future data stream. In contrast, VCRL is inherently more flexible and dynamic. It does not require pre-processing or assumptions about the global data distribution. Instead, VCRL performs filtering based on the group variance within the current training batch. This allows VCRL to adaptively select high-value samples without the computational overhead of periodic global reordering or the need for pre-computed difficulty metrics.

## E  CODE BENCHMARK RESULTS

To validate the effectiveness of VCRL on tasks outside of mathematical reasoning, we also conduct experiments on code generation tasks. Specifically, we conduct code-based RL training based on the *ProRL-1.5B-v2* (Liu et al., 2025a) to broaden the generalization of VCRL in terms of model selection. Throughout the model evaluation, we allow the model to execute Python code based on the Sandbox Fusion[6] environment. We allow the model to generate and execute code to assist in completing tasks, building upon the benchmarks of AIME-2024 and AIME-2025. We also use the LiveCodeBench (Jain et al., 2025), for a total of three benchmarks to comprehensively evaluate the performance of the training methods in code generation. The quantitative results are summarized in Table 4. VCRL achieves state-of-the-art performance across all benchmarks with an average score of 42.98, significantly outperforming the base model (30.99) and surpassing the strongest baseline, GSPO (40.86). Notably, on LiveCodeBench, VCRL reaches 35.20, confirming that our VCRL effectively generalizes to code generation tasks.

## F  SENSITIVITY ANALYSIS

To better illustrate how the hyperparameters in VCRL affect model performance, we present the results of the sensitivity analysis experiments shown in the Table 5 and 6. Specifically, Table 5 shows the sensitivity analysis of the threshold parameter $\kappa$ in variance-based dynamic sampling

---

[6] https://bytedance.github.io/SandboxFusion

Table 4: Performance comparison on code generation benchmarks using *ProRL-1.5B-v2*.

| Method | AIME-2024 w Code | AIME-2025 w Code | LiveCodeBench | Avg. |
|--------|------------------|------------------|---------------|------|
| Base Model | 40.00 | 27.50 | 25.48 | 30.99 |
| + GRPO | 50.62 | 33.95 | 31.11 | 38.56 |
| + DAPO | 52.50 | 34.17 | 30.79 | 39.15 |
| + GSPO | 53.13 | 36.88 | 32.58 | 40.86 |
| + VCRL | **54.79** | **38.96** | **35.20** | **42.98** |

Table 5: Sensitivity analysis based on the threshold parameter $\kappa$ in variance-based dynamic sampling in VCRL using *Qwen3-8B-Base*.

| VCRL $\kappa$ | AIME-2024 | AIME-2025 | MATH500 | OlympiadBench | AMC23 | Avg. |
|---------------|-----------|-----------|---------|---------------|-------|------|
| 0.4 | 30.42 | 26.88 | 92.00 | 60.55 | 68.98 | 55.77 |
| 0.6 | 25.00 | 23.12 | 90.71 | 57.55 | 64.91 | 52.26 |
| 0.7 | 30.42 | **27.29** | **92.78** | **61.94** | 69.73 | 56.43 |
| 0.8 | 34.38 | 27.08 | 91.99 | 60.21 | **75.15** | **57.76** |
| 0.9 | **32.08** | 26.25 | 92.68 | 60.92 | 71.69 | 56.72 |
| 0.95 | 28.75 | 26.25 | 92.36 | 60.73 | 70.41 | 55.70 |

of VCRL, and Table 6 shows the sensitivity analysis of the rollout group size $G$ in VCRL. Both experiments are conducted based on *Qwen3-8B-Base*.

Table 5 illustrates the impact of the variance-based dynamic sampling threshold $\kappa$ on VCRL performance. The results exhibit a clear trend where neither excessively small nor large values of $\kappa$ yield optimal outcomes. Specifically, when $\kappa$ is set to a low value, the average performance will decrease, which may be due to overly lenient sampling, retaining too many samples that may not be of use. Conversely, setting $\kappa$ too high also leads to a performance decline, as the mechanism may sample out too many samples with high variance. The best overall performance is achieved at $\kappa = 0.8$, which strikes an optimal balance, delivering the highest average score of 57.76 and demonstrating robust performance across diverse benchmarks, particularly on AMC23.

Table 6 presents the impact of the rollout group size $G$ in VCRL. In contrast to the threshold parameter $\kappa$, we observe a positive correlation between the rollout group size $G$ and model performance. Increasing $G$ consistently leads to better results across all benchmarks in average. Specifically, enlarging the group size from 4 to 16 yields a substantial performance boost, raising the average score from 44.17 to 57.76. This significant improvement indicates that a sufficient number of rollouts is essential for VCRL to accurately estimate the group variance, thereby ensuring the reliability of the variance-based dynamic sampling. While the best average performance (59.25) is achieved at the largest setting ($G = 32$), the marginal gains begin to diminish after $G = 16$ (improving by only 1.49 despite doubling the computational cost). This suggests that $G = 16$ already provides a robust estimation for effective learning, balancing performance with computational efficiency.

## G    COMPUTATION COST ANALYSIS

In VCRL, we introduce two key components: variance-based dynamic sampling and replay learning. In replay learning, we use a memory bank $\mathcal{M}$ to maintain and supplement high-$p$ training samples to improve training efficiency, but this also introduces potential additional computational cost. To compare the computational cost of VCRL with other benchmark methods in terms of actual computation time, we list the specific GPU computation times for the *Qwen3-8B-Base* experiments in Table 1, as shown in Table 7.

As shown in Table 7, the introduction of the replay learning in VCRL incurs a computational overhead, increasing the total training time for *Qwen3-8B-Base* from 90.72 hours (GRPO) to 112.01 hours—an increase of approximately 23.47%. However, a direct comparison of training steps does not fully account for this time difference. To evaluate performance under a fixed computational budget, we align the methods based on wall-clock time: approximately 380 steps of VCRL training

Table 6: Sensitivity analysis based on the rollout group size $G$ in VCRL using *Qwen3-8B-Base*.

| VCRL $G$ | AIME-2024 | AIME-2025 | MATH500 | OlympiadBench | AMC23 | Avg. |
|---|---|---|---|---|---|---|
| 4 | 18.95 | 17.29 | 82.71 | 47.75 | 54.14 | 44.17 |
| 8 | 23.54 | 18.75 | 85.21 | 49.49 | 58.06 | 47.01 |
| 12 | 28.54 | 26.67 | 87.77 | 55.48 | 65.06 | 52.70 |
| 16 | 34.38 | 27.08 | 91.99 | 60.21 | 75.15 | 57.76 |
| 24 | 34.38 | 27.50 | **93.74** | **60.90** | **77.86** | 58.88 |
| 32 | **35.83** | **29.79** | 93.46 | 60.88 | 76.28 | **59.25** |

Table 7: Comparison of training time using *Qwen3-8B-Base*. We report the total wall-clock time (in hours) required for the training process.

| Method | Training Time (hours) |
|---|---|
| GRPO | 90.72 |
| DAPO | 48.68 |
| GSPO | 89.53 |
| VCRL | 112.01 |

correspond to 500 steps of GRPO training. Referring to the training trajectories in Figure 3, we observe that even at this earlier stage (380 steps), VCRL remains highly competitive and outperforms GRPO trained for the full 500 steps. This demonstrates that although VCRL has a higher per-step cost, it achieves greater sample efficiency and utilizes the computational budget more effectively.

Table 8 details the changes in the effective batch size after variance-based dynamic sampling throughout the training process. In the early stages (e.g., steps 0–50), the mean batch size is relatively small (48–56), indicating that the variance-based sampling mechanism is actively removing a significant portion of samples to mitigate instability. As the policy optimizes and training stabilizes, the effective batch size progressively increases, reaching 128 in the final stages (steps 451–500). This trend demonstrates that as the training matures, the dynamic sampling mechanism effectively fades out, allowing the model to utilize the full batch of data when the variance-based selection is no longer necessary.

Table 8: Evolution of the effective mean batch size after variance-based dynamic sampling across training steps in VCRL for *Qwen3-8B-Base*.

| Step Interval | Mean Batch Size |
|---|---|
| 0–20 | 48 |
| 21–50 | 56 |
| 51–100 | 64 |
| 101–150 | 64 |
| 151–200 | 80 |
| 201–250 | 96 |
| 251–300 | 104 |
| 301–350 | 112 |
| 351–400 | 120 |
| 451–500 | 128 |

