# OpenReview forum: "VCRL: Variance-based Curriculum Reinforcement Learning for Large Language Models"
_ICLR.cc/2026/Conference — Submitted to ICLR 2026_

### Official Review · Reviewer_CFiz · 2025-10-28

**Soundness:** 3
**Presentation:** 3
**Contribution:** 3
**Rating:** 6
**Confidence:** 4

**Summary:**

This paper introduces Variance-based Curriculum Reinforcement Learning (VCRL), a novel framework designed to improve the training efficiency of Large Language Models (LLMs) with policy-based reinforcement learning. The central claim is that existing rollout-based RL methods, like GRPO, are suboptimal because they do not consider the difficulty of training samples relative to the model's current capabilities. The authors propose using the variance of rewards from multiple rollouts of a single prompt as a dynamic measure of its difficulty. The core hypothesis is that samples with high variance are the most valuable for learning. The authors conduct experiments on several mathematical reasoning benchmarks with Qwen3 models, demonstrating that VCRL outperforms strong RL baselines.

**Strengths:**

1. The concept of using group reward variance as a dynamic difficulty metric is intuitive. It provides a way to implement curriculum learning without needing pre-defined difficulty labels.
2. VCRL shows consistent state-of-the-art performance across five different mathematical reasoning benchmarks and two model sizes, outperforming strong baselines like GRPO, DAPO, and GSPO.
3. The paper provides a theoretical argument that VCRL's training should be more stable than GRPO's from a policy gradient norm perspective.

**Weaknesses:**

1. While the method is framed as "curriculum learning," the paper does not provide any direct visualization of this curriculum.
2. All experiments are conducted on mathematical reasoning tasks, which typically have a binary reward structure.

**Questions:**

1. To strengthen the "curriculum" claim, could you provide a visualization showing how the curriculum in the training batch evolves?
2. The paper's introduction claims that existing methods "fail to explicitly consider LLMs' learning ability for samples of different difficulty levels". Could you provide more direct evidence for this?

---

> ### Author Response · Authors · 2025-11-17
>
> We thank the reviewer for the careful reading and constructive feedback.
>
> ---
> **W1 & Q1**
>
> We give the training dynamics in Appendix A (Figure 4) and further clarified here: the group reward variance $p$, as a real-time, model-dependent indicator of difficulty, naturally forms the curriculum during training. In Figure 5 (Appendix B), we show how $p$ peaks when approximately half of the rollouts are successful—indicating that the samples are at the edge of the model's current capabilities, i.e., the optimal learning objective.
>
> In Figures 4a/4b (Appendix A), we plot the moving average and standard deviation of the reward scores for VCRL and GRPO over training steps. We observe that under VCRL, the reward increases steadily, especially in the early stages of training (e.g., the first 200 steps), while GRPO exhibits larger fluctuations. This steady improvement trend suggests that VCRL gradually exposes the model to increasingly difficult samples—a hallmark of curriculum learning.
>
> Furthermore, Figure 6 (Appendix C) shows that VCRL maintains a smaller and more stable gradient norm throughout the training process. This indicates that the optimization process avoids large-scale destructive updates caused by overly difficult or simple samples—consistent with the well-structured curriculum learned stepwise by the model.
>
> While we do not pre-sort samples by difficulty, VCRL dynamically constructs a curriculum based on the model’s instantaneous capability, filtering out samples that are too easy ($p$ low due to all correct) or too hard ($p$ low due to all wrong), and focusing on those with moderate difficulty (high $p$). We believe this adaptive mechanism better reflects true curriculum learning than static sorting.
>
> ---
> **W2**
>
> Although we used a binary reward distribution as an example to calculate the group reward for $k$ rollouts in this paper, VCRL is not limited to binary reward tasks. In scenarios with clearly defined rewards (continuous rewards or tiered rewards), we can also use the same calculation method for evaluation, only needing to modify the estimation of the variance bound in the method.
>
> ---
> **Q2**
>
> We argue that baseline methods like GRPO, DAPO, and GSPO do not explicitly account for the model’s changing ability relative to sample difficulty, even if they use rollout-based signals. In short, while prior work improves training mechanics (e.g., clipping, advantage estimation), none explicitly identify and select samples based on their suitability for the model’s current learning stage — which is exactly what VCRL achieves via variance-based dynamic sampling.
>
> ---
>
> We will upload the revised manuscript shortly after finalizing all responses. Should you have any additional questions or require further clarification, we would be pleased to provide it. Thank you once again for your insightful and constructive comments.

---

> > ### Comment · Reviewer_CFiz · 2025-11-21
> >
> > Thanks to the authors for the response. The response has addressed part of my questions and concerns. I would keep my score.

---

> > > ### Author Response · Authors · 2025-11-27
> > >
> > > We sincerely thank you for your follow-up response and for engaging in the discussion. We are glad to hear that our rebuttal has addressed your concerns. We truly appreciate your time, constructive feedback, and your positive assessment of our work. We will continue to polish the final version of the manuscript based on your helpful suggestions to ensure the highest quality.

---

### Official Review · Reviewer_pPKY · 2025-10-28

**Soundness:** 2
**Presentation:** 2
**Contribution:** 1
**Rating:** 2
**Confidence:** 4

**Summary:**

The paper proposes a variance-based curriculum reinforcement learning method to order the samples for training the mathematical reasoning LLMs. The method measures the difficulty of samples based on the group reward variance.

**Strengths:**

- The writing is easy to follow;
- The experiments look intensive;

**Weaknesses:**

- Curriculum RL (CRL) has been extensively studied [1], but the paper does not discuss the literature;
- Due to the lack of CRL literature, the baseline does not include any CRL method for comparison;
- Aware of the CRL literature, the proposed variance-based curriculum RL is not novel or interesting;
- The abstract includes too many acronyms.

[1] Portelas, Rémy, et al. "Automatic curriculum learning for deep rl: A short survey." arXiv preprint arXiv:2003.04664 (2020).

**Questions:**

See the weaknesses.

---

> ### Author Response · Authors · 2025-11-17
>
> We sincerely appreciate your valuable comments, which are very helpful to improve our work. We address your concerns as follows.
>
> - W1：The Curriculum RL (CRL) in the paper [1] refers to a broader field, primarily focusing on traditional Deep Reinforcement Learning, rather than the LLM RL field that VCRL focuses on. Many of the methods cited in the survey are not specifically designed for current language generation models and related RLVR tasks. This poses considerable difficulties for discussions on consistency.
> - W2 & W3：In our Response to W1 to Reviewer 6HFJ, we introduced a comparison with the ADCL, ADCL & EGSR [2] methods, where our method still demonstrates competitive results. At the same time, we also compared our method with many LLM CRL methods [3,4,5,6,7] to better illustrate the core and position of VCRL. Similarly, in Line 453-465 of our paper, we also discuss a great deal of recent works on LLM CRL to better illustrate the contributions and innovations of VCRLs to the field.
> - W4: We appreciate your feedback regarding the acronyms in the Abstract.
>
> We will be incorporating these suggestions into an upcoming version.
>
> [1] Portelas, Rémy, et al. "Automatic curriculum learning for deep RL: a short survey." Proceedings of the Twenty-Ninth International Conference on International Joint Conferences on Artificial Intelligence. 2021.
>
> [2] Zhang, Enci, et al. "Learning like humans: Advancing llm reasoning capabilities via adaptive difficulty curriculum learning and expert-guided self-reformulation." Proceedings of the 2025 Conference on Empirical Methods in Natural Language Processing. 2025.
>
> [3] Tzannetos, Georgios, et al. "Proximal curriculum for reinforcement learning agents." arXiv preprint arXiv:2304.12877 (2023).
>
> [4] Parashar, Shubham, et al. "Curriculum Reinforcement Learning from Easy to Hard Tasks Improves LLM Reasoning." arXiv preprint arXiv:2506.06632 (2025).
>
> [5] Chen, Xiaoyin, et al. "Self-Evolving Curriculum for LLM Reasoning." arXiv preprint arXiv:2505.14970 (2025).
>
> [6] Wang, Zhenting, et al. "Dump: Automated distribution-level curriculum learning for rl-based llm post-training." arXiv preprint arXiv:2504.09710 (2025).
>
> [7] Bae, Sanghwan, et al. "Online difficulty filtering for reasoning oriented reinforcement learning." arXiv preprint arXiv:2504.03380 (2025).

---

### Official Review · Reviewer_jpnt · 2025-11-01

**Soundness:** 2
**Presentation:** 2
**Contribution:** 3
**Rating:** 6
**Confidence:** 3

**Summary:**

This paper aims at improving the mathematical reasoning ability of LLMs with a curriculum learning framework. The proposed method is based on a simple but interesting observation: under the sparse reward setting, the agent's performance preserve low variance on both easy and hard tasks, while high with medium difficulty ones. Hence, the appropriate difficulty of experiences are determined as those with high group reward variance (Variance-based Dynamic Sampling). This work also proposes to use the high-variance samples to compose the experience replay (Replay Learning). The experiments are relatively extensive, conducted on five mathematical benchmarks with two variations of *Qwen3* as the base models.

**Strengths:**

The idea is conceptually simple but useful. The description of the method is for the most parts clear and the motivation is well explained. The empirical results on mathematical problem solving are promising. Compared to the reviewed related works, the variance-based criterion is simpler than those that requires an extra learnable module, which is easier to implement and potentially generalize to other tasks.

**Weaknesses:**

1. The work does not compare to other curriculum learning methodologies (e.g., Those mentioned in the related works: Hammoud et al. 2025, Feng et al. 2025a, etc.).

2. There is a mismatch between the claimed contributions and experimental setting. At the end of the introduction, the claim of contribution is quite general, indicating that the framework can be used for general RL tasks. However, the previous introduction and latter experiments both only focus on mathematical reasoning with LLMs. If the framework is claimed to be generalizable, other experiments should be conducted; Otherwise, the extension to other tasks should be held for merely a discussion, and the scope in the claim of contribution should be narrowed down.

**Questions:**

* **TECHNICAL QUESTIONS:**

1. The proposed framework is rather generalizable, with more potential with other tasks than merely mathematics. Do the authors have any insight or further results on other language tasks?

2. Is it possible to evaluate the token efficiency with the proposed method? If I understand correctly, after using the Replay Learning scheme proposed in this work, the RL can be more efficient. It would be interesting to evaluate that.

* **PRESENTATION SUGGESTIONS:**

1. Line 17, the acronym 'RLVR' was used before properly introducing the full term.

 2. Line 98, A verb is missing here. For example, 'a value model *estimated/calculated* by GAE' is better.

3. Line 132: 'clipping range' is better as 'clipping bound'?

4. If I understand correctly, the two points made in section 3.1 are conceptually the same thing: without curriculum learning, the current model parameter mismatches current training sample difficulty, rendering efficient learning difficult. The motivation to introduce curriculum learning is good and well-delivered from the text, but the writing can be more concrete.

5. The description around equation 7 can be clearer. Specifically, it is better to mention that the difference of the probabilities are $\\epsilon-$bounded, because $\\epsilon$ may also indicate a random noise or other meanings.

6. The authors should give a formality description of what a group of training samples that pertains to the same query is mathematically denoted.

7. Line 216: it is better to rephrase as 'the estimator achieves the maximum value of'

8. In Equation 12, should the $\\sigma\_{i,max}²$ be $\\sigma\_{max}²$, because it is the group-wise maximum of variance?

---

> ### Author Response · Authors · 2025-11-17
>
> We are very grateful for your recognition of our work and your detailed suggestions. Regarding your technical questions:
>
> - TQ1: VCRL is a task-independent approach, as long as there is a verifiable reward signal. Based on this, we presented our performance on coding tasks in our Response to W2 for Reviewer 6HFJ, which was also quite competitive.
> - TQ2: Your understanding is correct. Compared to GRPO, VCRL does indeed perform better in terms of training efficiency, which is easily reflected in the training dynamics in Figure 2 and 3. Specifically, VCRL requires fewer training steps to achieve the same model performance compared to other baseline methods.
>
> We also appreciate your suggestions regarding the presentation. We will be incorporating these suggestions into an upcoming version.

---

### Official Review · Reviewer_6HFJ · 2025-11-02

**Soundness:** 3
**Presentation:** 2
**Contribution:** 1
**Rating:** 4
**Confidence:** 4

**Summary:**

This paper proposes a curriculum strategy for reinforcement learning that prioritizes training examples based on how inconsistent the model’s rewards are across multiple rollouts of the same sample. High-variance problems are treated as most informative because the model sometimes succeeds and sometimes fails on them, indicating useful learning signal. The method filters for such problems at each step, supplements batches with a replay buffer that stores previously identified high-value items, and keeps the underlying GRPO-style optimization unchanged. Experiments on several math reasoning benchmarks with Qwen3 models demonstrate the effectiveness of the proposed method.

**Strengths:**

1. The studied problem is important. This paper is well-motivated.

2. The theoretical analysis and gradient-norm plots suggest the proposed method has smoother updates than GRPO are provided.

3. Different LLM RL methods are included in the experiments, showing the generalizability of the proposed method.

**Weaknesses:**

1. The biggest problem is that this paper does not compare against existing curriculum-learning methods for (LLM) reinforcement learning, despite a growing body of existing works [1,2,3,4,5,6]. The reported baselines are non-curriculum algorithms (e.g., vanilla GRPO), which makes it hard to compare the proposed variance-based curriculum signal and other signals. In addition, the related-work section does not adequately position the method within prior curriculum strategies on LLM RL. A better evaluation should includes more existing RL curriculum learning methods. It is also suggested to expand the related-work discussion to clarify what is new versus known in curriculum design for (LLM) RL. The high level idea that uses a signal to represent the learning difficulties and design the curriculum based on the difficulties is not new and are widely used in the listed existing works. It might need more discussion to justify the novelty and the superiority of the variance based signals. What is the fundamental differences and advantages comparing to these existing works?

2. The experiments use only Qwen models on math reasoning benchmarks. Although Qwen is widely-used open-sourced model family, recent works point out that it may have exposure to some math benchmark data [7], which could influence results. Please broaden coverage to additional domains (e.g., coding) and include more model families to make the empirical results more comprehensive.

3. The method appears to discard some rolled-out samples and then fetch replacements from a memory bank, which likely increases per-step compute and latency relative to GRPO. The paper does not quantify this cost. Please report wall-clock time to a target accuracy comparing to baselines, the ratio of discarded to reused rollouts, sensitivity to the number of rollouts and to the difficulty threshold, and a breakdown of memory-bank overhead. Efficiency claims should be supported under the total compute/time cost, rather than only step numbers or fixed-step comparisons.


[1] Zhang et al., Learning Like Humans: Advancing LLM Reasoning Capabilities via Adaptive Difficulty Curriculum Learning and Expert-Guided Self-Reformulation. EMNLP 2025.

[2] Tzannetos et al., Proximal Curriculum for Reinforcement Learning Agents

[3] Parashar et al., Curriculum Reinforcement Learning from Easy to Hard Tasks Improves LLM Reasoning

[4] Chen et al., Self-Evolving Curriculum for LLM Reasoning

[5] Wang et al., DUMP: Automated Distribution-Level Curriculum Learning for RL-based LLM Post-training

[6] Bae et al., Online Difficulty Filtering for Reasoning Oriented Reinforcement Learning

[7] Wu et al., Reasoning or Memorization? Unreliable Results of Reinforcement Learning Due to Data Contamination

**Questions:**

See above.

---

> ### Author Response · Authors · 2025-11-14
> **Response to W1**
>
> We sincerely appreciate your valuable comments, which are very helpful to improve our work. We address your concerns as follows.
>
> We greatly appreciate your addition to the research on curriculum reinforcement learning methods. We acknowledge that the paper does indeed lack comparisons with relevant baseline methods, especially in the LLM RL field. Based on the relevant work you pointed out, we selected ADCL [1] as a comparable baseline method and present our performance in the experimental settings aligned with this ADCL method in the table below. VCRL shows competitive results. Specifically, we maintain consistency with ADCL, training Qwen2.5-7B, generating 8 rollouts for each question, using a batch size of 1024, a fixed learning rate of 1e-6, a maximum response length of 4096, and a temperature parameter of 0.7.
>
> | Qwen2.5-7B | AIME-2024 | AIME-2025 | MATH500 | AMC23 | Avg. |
> |:---|---:|---:|---:|---:|---:|
> | Base | 16.67 | 16.67 | 69.40 | 32.50 | 33.81 |
> | GRPO | 26.67 | 16.67 | 72.40 | 45.00 | 40.19 |
> | ADCL | 33.33 | 30.00 | 76.20 | 55.00 | 48.63 |
> | ADCL & EGSR | 36.67 | 33.33 | 81.80 | 55.00 | 51.70 |
> | VCRL | 35.83 | 32.71 | 79.73 | 61.22 | 52.37 |
>
> Furthermore, ADCL performs difficulty-based reordering on the training data (e.g., reordering every 100 training steps), which is the core method of ACRL proposed in this paper. In comparison, VCRL is more flexible and dynamic. VCRL does not operate on or make assumptions about future training data stream, but rather filters based on the group variance of the data within the current training batch.
>
> For [2], this work does not primarily focus on the field of LLM RL, but rather on the derivation of the theoretical foundation of the method. For [3, 4], they all have a division and preset of task difficulty levels, which is different from the dynamic characteristics of VCRL. For [5], DUMP requires a complete understanding of the task's difficulty distribution. If a distribution (e.g., "K & K 7 characters") is selected as high priority by DUMP's UCB algorithm, then the probability of selecting all samples (regardless of difficulty) from that distribution will increase. VCRL, on the other hand, does not have such strict requirements and restrictions on tasks. For [6], ODF employs a pass-based selection strategy, similar to DAPO, continuously sampling data to obtain training batches. VCRL, on the other hand, introduces Memory Bank Replay, repeatedly learning from high-quality past data to reduce this cost and improve training efficiency.
>
> We acknowledge once again the lack of comparison in this area, and we are conducting experiments to introduce more baseline method comparisons. We hope that the above comparisons and discussions will enhance the presentation of the paper and better clarify the position of our work.
>
> [1] Zhang, Enci, et al. "Learning like humans: Advancing llm reasoning capabilities via adaptive difficulty curriculum learning and expert-guided self-reformulation." Proceedings of the 2025 Conference on Empirical Methods in Natural Language Processing. 2025.
>
> [2] Tzannetos, Georgios, et al. "Proximal curriculum for reinforcement learning agents." arXiv preprint arXiv:2304.12877 (2023).
>
> [3] Parashar, Shubham, et al. "Curriculum Reinforcement Learning from Easy to Hard Tasks Improves LLM Reasoning." arXiv preprint arXiv:2506.06632 (2025).
>
> [4] Chen, Xiaoyin, et al. "Self-Evolving Curriculum for LLM Reasoning." arXiv preprint arXiv:2505.14970 (2025).
>
> [5] Wang, Zhenting, et al. "Dump: Automated distribution-level curriculum learning for rl-based llm post-training." arXiv preprint arXiv:2504.09710 (2025).
>
> [6] Bae, Sanghwan, et al. "Online difficulty filtering for reasoning oriented reinforcement learning." arXiv preprint arXiv:2504.03380 (2025).

---

> ### Author Response · Authors · 2025-11-14
> **Response to W2**
>
> We fully understand your concerns regarding the Qwen series models, especially the issue of data exposure [7]. Therefore, we have introduced additional experimental results as shown in the table below. Among them, ProRL-1.5B-v2 [8] is a recently developed inference model. Furthermore, our evaluations of AIME-2024 and AIME-2025 allow the model to invoke code to assist in solving problems.
>
> | ProRL-1.5B-v2 | AIME-2024 w Code | AIME-2025 w Code | LiveCodeBench | Avg. |
> |:---|---:|---:|---:|---:|
> | base | 40.00 | 27.50 | 25.48 | 30.99 |
> | GRPO | 50.62 | 33.95 | 31.11 | 38.56 |
> | VCRL | 54.79 | 38.96 | 35.20 | 42.98 |
>
> In addition, we are experimenting with more models to improve the generalization of VCRL.
>
> [7] Wu, Mingqi, et al. "Reasoning or memorization? unreliable results of reinforcement learning due to data contamination." arXiv preprint arXiv:2507.10532 (2025).
>
> [8] Liu, Mingjie, et al. "Prorl: Prolonged reinforcement learning expands reasoning boundaries in large language models." arXiv preprint arXiv:2505.24864 (2025).

---

> ### Author Response · Authors · 2025-11-14
> **Response to W3**
>
> We acknowledge the lack of cost-related discussions in our work regarding memory bank and fully understand your concerns in this area. Based on your comments, we have introduced comparisons of actual training time, changes in sample size after dynamic sampling during training, sensitivity experiments to rollouts and sampling thresholds, and a practical analysis and discussion of memory bank overhead.
>
> | Qwen3-8B-Base | Training Time (hours) |
> |:---|---:|
> | GRPO | 90.72 |
> | DAPO | 48.68 |
> | GSPO | 89.53 |
> | VCRL | 112.01 |
>
> As we demonstrated in our training time comparison, after introducing memory bank replay, VCRL increased the training time for Qwen3-8B-Base from 90.72 hours to 112.01 hours compared to the basic GRPO, representing a 23.47% increase. If we change the comparison from being based on the number of training steps to being based on computational cost, approximately 380 steps of VCRL training is comparable to 500 steps of GRPO training. According to the results shown in Figure 3, even at around 380 steps, our proposed VCRL remains highly competitive compared to 500 steps of GRPO.
>
> ---
>
> | Step | Mean Batchsize after Dynamic Sampling |
> |:---|---:|
> | 0-20 | 48 |
> | 21-50 | 56 |
> | 51-100 | 64 |
> | 101-150 | 64 |
> | 151-200 | 80 |
> | 201-250 | 96 |
> | 251-300 | 104 |
> | 301-350 | 112 |
> | 351-400 | 120 |
> | 451-500 | 128 |
>
> ---
>
> | Qwen3-8B-Base $\kappa$ | AIME-2024 | AIME-2025 | MATH500 | OlympiadBench | AMC23 | Avg. |
> |:---|---:|---:|---:|---:|---:|---:|
> | 0.4 | 30.42 | 26.88 | 92.00 | 60.55 | 68.98 | 55.77 |
> | 0.6 | 25.00 | 23.12 | 90.71 | 57.55 | 64.91 | 52.26 |
> | 0.7 | 30.42 | 27.29 | 92.78 | 61.94 | 69.73 | 56.43 |
> | 0.8 | 34.38 | 27.08 | 91.99 | 60.21 | 75.15 | 57.76 |
> | 0.9 | 32.08 | 26.25 | 92.68 | 60.92 | 71.69 | 56.72 |
> | 0.95 | 28.75 | 26.25 | 92.36 | 60.73 | 70.41 | 55.70 |
>
> ---
>
> | Qwen3-8B-Base rollout | AIME-2024 | AIME-2025 | MATH500 | OlympiadBench | AMC23 | Avg. |
> |:---|---:|---:|---:|---:|---:|---:|
> | 4 | 18.95 | 17.29 | 82.71 | 47.75 | 54.14 | 44.17 |
> | 8 | 23.54 | 18.75 | 85.21 | 49.49 | 58.06 | 47.01 |
> | 12 | 28.54 | 26.67 | 87.77 | 55.48 | 65.06 | 52.70 |
> | 16 | 34.38 | 27.08 | 91.99 | 60.21 | 75.15 | 57.76 |
> | 24 | 34.38 | 27.50 | 93.74 | 60.90 | 77.86 | 58.88 |
> | 32 | 35.83 | 29.79 | 93.46 | 60.88 | 76.28 | 59.25 |

---

> ### Author Response · Authors · 2025-11-14
>
> We will upload the revised manuscript shortly after finalizing all responses. Should you have any additional questions or require further clarification, we would be pleased to provide it. Thank you once again for your insightful and constructive comments.

---

> ### Author Response · Authors · 2025-11-26
> **Gentle Request for Discussion**
>
> Dear Reviewer 6HFJ,
>
> We are sorry to bother you via the comment thread. Since the rebuttal period is already halfway through and we have not yet received your feedback, we wanted to gently follow up.
>
> We highly value your comments and have carefully responded to each of the Weaknesses you raised, including adding corresponding experiments and analyses. We would be very grateful if you could take a look when you have time, and we sincerely hope our responses can address your concerns. If you have any additional questions or remaining doubts, we would be more than happy to use the remaining rebuttal time to provide further clarification or engage in additional discussion.
>
> Again, we apologize for the interruption and thank you very much for your time.
>
> Best regards,
>
> The authors

---

### Author Response · Authors · 2025-11-25
**Summary of Revised Manuscript**

Based on the reviewers' comments, we have uploaded an updated manuscript. Compared to the previous version, we have made the following major changes:

- In response to the lack of comparison with related work mentioned by Reviewer 6HFJ, we have strengthened the discussion in the Related Work (Line 451-471) and added completed experiments to Appendix D to further illustrate the position of VCRL in current LLM RL methods.
- To address the issues of task domain and model coverage mentioned in Reviewer 6HFJ and Reviewer CFiz, we add Appendix E, which uses the ProRL-1.5B-v2 model to evaluate the performance of VCRL on the code generation task compared to various training baseline methods, in order to better verify the generalization ability of VCRL on the model and task.
- In response to the lack of sensitivity analysis for some key parameters in VCRL mentioned in Reviewer 6HFJ, we have added Appendix F to supplement the sensitivity analysis experiments and corresponding discussions for the sampling threshold and rollout group size in VCRL.
- To address the lack of discussion regarding the potential computational costs introduced in VCRL, as mentioned in Reviewer 6HFJ, we add Appendix G, which delves into this issue in depth, examining the impact of actual training time and dynamic sampling on batch size. Even with the same training time, VCRL's average performance across various benchmarks still outperforms the compared baseline methods.
- Fix some presentation issues, and we're very grateful for Reviewer jpnt's suggestions!

We would like to thank the reviewers for their insightful comments and constructive suggestions again, which significantly improved the quality and presentation of our paper. We hope that our extensive new experiments and detailed clarifications have satisfactorily addressed your concerns. We would greatly appreciate it if you could reconsider your evaluation in light of these improvements, and we remain open to any further discussions.

---

### Meta-Review · Area_Chair_GSXP · 2026-01-08

**Summary:**

The reviewers appreciate the practical utility of the paper in designing an algorithm to train reasoning models

**Reviewer Concerns:**

The reviewer's main concern relate to the lack of extensive baseline comparison in the extensive literature of curriculum learning for reasoning models. Several baselines are not implemented. The performance comparison is at the per RL step level, but each RL steps costs more. Compute-normalized performance is not reported.

**Reviewer Scores:**

Unsure if reviewers would change their score. Rev 1 has concerns about e.g. compute cost which is not fully addressed. Rev 3 rejects but without much justification, I guess they would be unwilling to raise their score. Other revs are already positive, and rebuttal to those is accordingly limited.

---

### Decision · Program_Chairs · 2026-01-26

Reject